# Reliable Detection of Atrial Fibrillation with a Medical Wearable during Inpatient Conditions

**DOI:** 10.3390/s20195517

**Published:** 2020-09-26

**Authors:** Malte Jacobsen, Till A. Dembek, Athanasios-Panagiotis Ziakos, Rahil Gholamipoor, Guido Kobbe, Markus Kollmann, Christopher Blum, Dirk Müller-Wieland, Andreas Napp, Lutz Heinemann, Nikolas Deubner, Nikolaus Marx, Stefan Isenmann, Melchior Seyfarth

**Affiliations:** 1Faculty of Health, University Witten/Herdecke, 58448 Witten, Germany; athanasios-panagiotis.ziakos@helios-gesundheit.de (A.-P.Z.); nikolas.deubner@helios-gesundheit.de (N.D.); stefan.isenmann@st-josef-moers.de (S.I.); melchior.seyfarth@helios-gesundheit.de (M.S.); 2Department of Internal Medicine I, University Hospital Aachen, RWTH Aachen University, 52074 Aachen, Germany; dirmueller@ukaachen.de (D.M.-W.); anapp@ukaachen.de (A.N.); nmarx@ukaachen.de (N.M.); 3Department of Neurology, Faculty of Medicine, University of Cologne, 50937 Cologne, Germany; till.dembek@uk-koeln.de; 4Department of Cardiology, Helios University Hospital of Wuppertal, 42117 Wuppertal, Germany; 5Department of Computer Science, Heinrich Heine University Düsseldorf, 40225 Düsseldorf, Germany; rahil.gholamipoorfard@hhu.de; 6Department of Hematology, Oncology, and Clinical Immunology, University Hospital Düsseldorf, Medical Faculty, Heinrich Heine University Düsseldorf, 40225 Düsseldorf, Germany; kobbe@med.uni-duesseldorf.de; 7Department of Biology, Heinrich Heine University Düsseldorf, 40225 Düsseldorf, Germany; markus.kollmann@hhu.de (M.K.); christopher.blum@hhu.de (C.B.); 8Science-Consulting in Diabetes, 41462 Neuss, Germany; l.heinemann@science-co.com; 9Department of Neurology, St. Josef Hospital, 47441 Moers, Germany

**Keywords:** clinical trial, wearable sensors, atrial fibrillation, photoplethysmography, deep neural network

## Abstract

Atrial fibrillation (AF) is the most common arrhythmia and has a major impact on morbidity and mortality; however, detection of asymptomatic AF is challenging. This study aims to evaluate the sensitivity and specificity of non-invasive AF detection by a medical wearable. In this observational trial, patients with AF admitted to a hospital carried the wearable and an ECG Holter (control) in parallel over a period of 24 h, while not in a physically restricted condition. The wearable with a tight-fit upper armband employs a photoplethysmography technology to determine pulse rates and inter-beat intervals. Different algorithms (including a deep neural network) were applied to five-minute periods photoplethysmography datasets for the detection of AF. A total of 2306 h of parallel recording time could be obtained in 102 patients; 1781 h (77.2%) were automatically interpretable by an algorithm. Sensitivity to detect AF was 95.2% and specificity 92.5% (area under the receiver operating characteristics curve (AUC) 0.97). Usage of deep neural network improved the sensitivity of AF detection by 0.8% (96.0%) and specificity by 6.5% (99.0%) (AUC 0.98). Detection of AF by means of a wearable is feasible in hospitalized but physically active patients. Employing a deep neural network enables reliable and continuous monitoring of AF.

## 1. Introduction

Atrial fibrillation (AF) is the most common arrhythmia with rising incidence and prevalence [1,2]; the current prevalence is estimated to be between 2% to 4% [3]. AF is more common in males and shows an increasing prevalence with age [4]. There are a number of modifiable known risk factors for AF, including obesity, hypertension, diabetes mellitus, and smoking, as possible contributors to the development and progression of AF [5].

AF is associated with a broad spectrum of clinical events, including ischemic stroke. The proportion of time in AF associated with a significant risk for complications is unknown, thus requiring further evaluation [6]. Due to the paroxysmal and often asymptomatic occurrence of AF, ECG Holter monitoring is frequently employed to detect episodes of silent AF [7]. However, ECG Holter monitoring has limitations: Carrying an ECG Holter limits patients in their daily activities and restricts monitoring to relatively short periods of time. Additionally, ECG Holters are prone to movement artifacts, and thus, not reliable during phases of physical activity [8]. 

Wearables that are used as medical devices (defined as having a regulatory approval like a Conformité Européenne (CE) mark for Europe) offer an affordable non-invasive screening option for AF [9,10,11,12,13]. Photoplethysmography (PPG) is frequently employed in such wearables [14]. It is an optical method to measure volume changes in the tissue. PPG is used to calculate clinically relevant parameters, e.g., heart rate, inter-beat intervals (IBI—the interval between two pulse waves in milliseconds) [15]. Intervals between heartbeats are a parameter often used for the detection of AF. PPG derived IBI show a high correlation to the ECG derived heart rate intervals (gold-standard) [16]. Technologies employed in wearables and evaluated for the detection of AF are most often based on single-lead ECG or PPG and can be separated into active and passive approaches: Active monitoring requires that the patient initialized a recording, e.g., individuals have to place their fingers on the electrodes of a smartphone like device. In contrast, wearables with a passive monitoring approach do not require patient intervention. With this approach, measurements are performed continuously or semi-continuously (e.g., every 5 min). In a previous clinical trial with an active approach, wearable detection of AF was possible with a sensitivity of 91.5% and specificity of 99.6% [13]. In clinical trials with a passive approach, equivalent results were shown in patients where physical activity was restricted while recording. However, there was a risk of missing asymptomatic episodes of AF. When such wearables are used for ECG recordings, usage of adhesives or bandages is needed, and there are limitations regarding diagnostic adherence [9]. In the Huawei Heart Study, more than one-third of individuals with suspected AF were primarily detected with a periodical passive PPG approach [12]. However, a recent trial using a passive approach showed that there is a gap in detecting AF under controlled and uncontrolled conditions, most likely due to periods of physical activity with an increase in heart rate and movement artifacts [17]. Some wearables under evaluation had varying sampling rates, with considerable risk of missing AF [10]. 

A novel upper arm medical wearable (Everion^®^, Biovotion AG, Switzerland) employs a passive PPG approach, allowing reliable long-term, high-resolution data recording. This device records of patients’ physical activities during recording and provides information about the proportion of the automatically interpretable time. 

The aim of this study was to evaluate the performance of a medical wearable by means of employing a PPG technology for AF detection in patients with paroxysmal or persistent AF study during inpatient conditions.

## 2. Materials and Methods

This study was an open-label, single-arm, inpatient, single-center trial. The clinical investigation plan was approved by the Ethical Committee of the University Witten/Herdecke, Germany, and was registered in the German clinical trials register (DRKS00014821). 

Patients were recruited consecutively at the Department of Cardiology, University Hospital Wuppertal between September and December 2018 (Figure 1). 

The primary outcome of this trial was the evaluation of sensitivity and specificity of non-invasive AF detection by a medical wearable at rest and during moderate physical activity. The secondary outcome was the determination of the proportion of recording time interpretable by algorithms.

All patients gave written informed consent prior to enrolment in this trial. Admitted patients with documented AF (e.g., prior to electrical cardioversion) or known paroxysmal AF were screened for eligibility for trial participation. Inclusion criteria were patients admitted for AF by their treating cardiologist and emergency room show ups with age ≥ 18 years and an indication for ECG Holter monitoring. Exclusion criteria were any cardiac implants or conditions which might impair measurements (e.g., upper arm tattoos, skin diseases).

Patients had no restrictions on their physical activity. At the end of the monitoring period, a safety assessment was performed. Patients answered a short questionnaire at the study end to evaluate wearable usage (discomfort, pain, sense of safety, design, willingness to perform inpatient, and outpatient monitoring).

In line with the standard of care in the hospital, patients carried a three-lead ECG Holter (Lifecard CF, Spacelabs Healthcare GmbH, Germany) for detection of AF over 24 h. ECG Holter data were reviewed for atrial arrhythmias by two cardiologists independently using a standard of care software tool (Sentinel 10, Spacelabs Healthcare, Snoqualmie, WA, USA). In the case of differing diagnoses, a third cardiologist was consulted. Heart rhythm was classified into either sinus rhythm, AF, or atrial flutter, and this classification served as the gold standard for further analysis (Figure 2). ECG datasets were discarded if more than 50% of recorded data was not interpretable as defined by our independent raters.

In parallel, a commercially available medical wearable (Everion, Biovotion AG, Switzerland) was worn by the patients. The wearable was attached to the preferred upper arm of the patients by the investigator. The time base of the wearable was synchronized to the ECG Holter. The wearable is a CE marked medium-risk device (class IIa), according to the Directive 93/42/EEC (firmware used was for clinical investigation only). It has different sensors for non-invasive monitoring of vital signs (e.g., PPG, accelerometry, gyroscope), memory storage of 16 MB Flash and a battery life of up to 32 h. Parameters, such as heart rate, IBI, the morphology of the pulse wave, and a physical activity index (based on the accelerometry data), are calculated using proprietary algorithms of the manufacturer implemented in the firmware. PPG-Signals were acquired with a sampling rate of 51.2 Hz. IBI were calculated permanently and stored approximately every 40 s. The device also provides recording quality indices for each data point. Data stored in the wearable were downloaded via a Bluetooth connection.

Two different approaches for detecting AF from the downloaded data were investigated: First, an established metric for AF detection, the normalized root mean square of successive differences (nRMSSD) of the IBIs, to differentiate between sinus rhythm and AF was used [11]. Second, a deep neural network (DNN) to detect episodes of AF was applied. Data with an insufficient quality based on the point-in-time accuracy estimate in the pre-processed data were excluded. Sufficient quality was defined when such an estimate for the IBI values could be calculated.

nRMSSD classification: Data was split into successive five-minute periods, and nRMSSD was calculated for all of these. For determining the optimal nRMSSD threshold, the dataset was split into a ‘training cohort’ consisting of the first 80% of the recruited patients and a ‘testing cohort’ consisting of the remaining 20%. Receiver operating characteristics (ROC) were calculated for five-minute periods in the ‘training cohort’, and the threshold with the highest Youden’s J statistic was determined. This threshold was then applied to calculate the sensitivity and specificity of nRMSSD based AF detection in the ‘testing cohort’. Algorithms presented were not trained to differentiate between AF and atrial flutter, only to discriminate AF.

Deep neural network classification: As data source, the same five-minute periods of IBI values were used as for the nRMSSD-model described above. As the dataset contained significantly more non-AF periods than AF periods, oversampling was performed by replicating the randomly selected samples to achieve a balanced dataset. The IBI values were encoded together with their associated quality scores into a multi-dimensional vector space, where IBI values with different quality scores are taken orthogonal to each other. A DNN was trained unsupervised on the dataset to extract the relevant features for AF detection. The training objective was given by maximizing the mutual information between IBI values that were separated by a randomly chosen time point within the five-minute period. The algorithmic details for computing of mutual information can be found in the appendix (see Appendix B) [18]. The unsupervised classification was carried out by one-nearest neighbor classification (Figure 3). Additionally, a second DNN (classifier) was trained on the extracted features from unsupervised learning using annotated data. The evaluation of the DNNs were carried out by randomly splitting the pre-processed data into the train (80%)/validation (10%)/test datasets (10%). Subsequently, sensitivity and specificity were calculated using ten-fold cross-validation. For testing, the unbalanced original data was used.

A prerequisite for reliable detection of AF over time in clinical practice is sufficient data quality and that the recording time is maximal, e.g., a given patient might carry a wearable for 24 h; however, the proportion of recording time automatically interpretable by algorithms (=interpretable time) may be decisively less. [19,20] From the data obtained, the percentage of ‘good quality data’ was assessed by aggregating the time periods during which data were available that enabled an automatic IBI analysis. Others have used a cut-off value of 90% analyzable data for each five-minute period in resting patients; however, in order to apply a pragmatic approach in potentially active patients value of 80% was used for this trial. A threshold of ≥ 80% of the interpretable time was considered to be sufficient for clinical monitoring. Logistic regression analysis was used to evaluate which factors have an impact on the analyzable time. To evaluate the success of wearable data recording, the total recording time, as well as total interpretable time (time with accepted quality indices), were calculated. 

Due to the known effect of patients’ physical activity on the detection of AF, an activity index over time was calculated for each patient. Based on the activity classification provided by the wearable, any classification besides ‘resting’ was considered as physical activity (e.g., walking flat). From the activity data provided by the wearable subsequent five-minute periods were labeled as ‘active’ or ‘resting’. The activity index is expressed as a percentage of each hour of recording. It was analyzed if detection of AF was possible with the wearable used during periods with and without physical activity. 

For accuracy testing of heart rate estimation by the wearable in patients with different underlying heart rhythms, in each patient, one hour of ECG recording with a low rate of artifacts was selected manually (see Appendix B). Accuracy evaluation was performed as described elsewhere [21]. For data analysis, a standard software tool was used (MATLAB R2018b; MathWorks, Natick, MA, USA). Statistical Analysis

The confidence interval was set to 95% for all statistical analyses. Non-parametric categorical distributed variables were tested with a 2-tailed Fishers exact test or Chi-Square test. Continuous variables were tested with the Mann-Whitney test. For analyses of variables that have an impact on interpretable time, logistic regression was performed. For the primary outcome of AF detection Receiver Operating Characteristics (ROC) analysis for nRMSSD was performed, and the area under the curve (AUC) of the ROC-analysis was calculated.

## 3. Results

Five of the 107 patients enrolled were excluded, due to missing data or poor ECG Holter data quality. The 102 patients analyzed (age 71.0 ± 11.9 years; 52% male) had a mean CHA_2_DS_2_-VASc-Scores of 2.7. Demographical data, comorbidities, and concomitant medication of these patients are given in Table 1. 

By means of ECG Holter recording the patients were diagnosed (Cohens kappa 0.87) as having: Only sinus rhythm (*n* = 43, 42.2%), AF (*n* = 48, 47.0%), or atrial flutter episodes (*n* = 11, 10.8%). Patients with sinus rhythm were younger compared to those with AF (*p* = 0.026). There were no significant differences between patients with different heart rhythms with respect to comorbidities and concomitant medication.

The mean data recording time was 23.0 ± 3.3 h, comprising 2306 h of total recording time. In 62 out of the 102 patients (60.8%), the interpretable time was ≥80%; for the algorithms applied 1781 h (77.2%; average of 17.7 h) were evaluable (Table 2 and Table 3); however, the time varied considerably among patients (SD 23.2%).

1nRMSSD-based algorithm

Detection of AF in the algorithm testing dataset was possible with a sensitivity of 95.2% and a specificity of 92.5% (Table 2) based on nRMSSD algorithm. Data obtained with the ECG Holter contained 5156 five-minute periods of AF. For 4469 of these episodes, simultaneous wearable data of sufficient quality was available. Of these 4,469 periods (algorithm training and algorithm testing), 4141 were correctly classified (true positive) as AF. In total, 1905 periods were classified false-positive, 328 periods were false-negative. Of the 1905 false-positive periods, 88 (4.6%) had a positive activity index. During 3,464 five-minute time periods with physical activity, AF was present in 755 (21.8%) periods. Of these, 701 periods were correctly classified as AF with a minor decrease in sensitivity (92.9%) and specificity (85.5%). 

2DNN-based algorithm

Further improvement in the detection of AF was achieved by means of the DNN (Table 2). On 10 different training /validation splits, the best model achieved a sensitivity of 96.9% and specificity of 95.4% (AUC 0.99). With ten-fold cross-validation of the models applied to the test set resulted in an average sensitivity of 96.9 ± 0.3% and a specificity of 95.0 ± 0.4% (AUC 0.99 ± 0.1). Applying a fully unsupervised approach to the complete datasets resulted in a sensitivity of 96.7% and a specificity of 98.6% (AUC 0.98). With the same cross-validation methods applied to the test set on average, a sensitivity of 96.0 ± 0.4% and a specificity of 99.0 ± 0.2% (AUC 0.98 ± 0.2) was achieved. In the five-minute periods with a positive physical activity index, sensitivity (96.8 ± 0.6), and specificity (96.9 ± 0.5; AUC 98.9 ± 0.1) of AF detection remained unchanged with the DNN.

3Further analysis

Patients with an interpretable time ≥ 80% were allocated to one group, and differences in comorbidities, concomitant medication, arm circumference, and activity index were compared to those with an interpretable time < 80% (Table 3). Descriptive characteristics between the two groups differed as follows: Patients with an interpretable time < 80% were older (*p* = 0.023) and used more antihypertensive agents (beta-blockers, *p* = 0.020; renin-angiotensin system inhibitors, *p* = 0.031). Logistic regression analysis showed that age (*p* = 0.039, OR 0.95, CI 0.904–0.997) had a negative impact on interpretable time. In contrast, height had a positive impact (*p* = 0.002, 1.10, 1.034–1.162). 

Measurement conditions in both groups with respect to heart rhythm, side of recording, arm circumference, and activity index were comparable (Table 3). The physical activity level of all patients during 24 h was 16.1% based on a positive activity index in five-minute periods.

The activity index showed peaks after breakfast and in the afternoon (Figure 4).

Carrying the wearable did not induce any discomfort or pain in 97.5% of the patients. More than 70% of the patients could envisage using such a wearable for home monitoring. No serious adverse effects were observed during the trial; however, one device-related adverse effect was observed; a skin irritation after wearing the device was fully reversible after six days.

## 4. Discussion 

Our study suggests that reliable detection of AF in high-risk patients for AF is possible with the medical wearable used, also during time periods with physical activity. The deep neural network approach showed an even better ability of AF detection than the established nRMSSD algorithm. The DNN approach enables a reliable computer-based analysis, and thereby, the option of a real-time AF detection. Using a passive measurement approach, a high interpretable time proportion (77.2%) was achieved.

The high-risk population studied was comparable with respect to age and cohort distribution in terms of heart rhythm to the population of the multicenter trial of Brasier et al. [13]; however, the population in their trial had a higher mean CHA_2_DS_2_-VASc-Scores reflecting a higher prevalence of comorbidities.

Detection of AF with nRMSSD in five-minute periods showed higher sensitivity, but lower specificity than in other studies conducted with an active measurement approach; however, our results were obtained in a not physically restricted population [11,13]. Detection of AF within periods of physical activity represents a challenge for wearables (also with ECG Holter monitoring). In some trials, there was a gap in the detection of AF during physical active vs. restricted physical conditions [17]. In other trials, like the Apple Heart study, no measurements were performed while participants were physically active [10]. In our trial, the overall physical activity index (as provided by the wearable) observed probably does not reflect real-world physical activity, since only inpatients were enrolled. Nevertheless, also in periods with a positive activity index, detection of AF was feasible with good reliability. However, it is a limitation that the activity index used was not assessed with a standardized reference method in parallel. There is a difference between the number of available five-minute periods in AF in ECG Holter and wearable data., i.e., due to the interpretable time of wearable data. 

The deep learning setup applied-consisting of unsupervised feature extraction followed by unsupervised classification-showed higher sensitivity and specificity in detecting AF. These results were comparable to Tison et al. [17]; additionally, they were achieved with unlabeled data. Large amounts of unlabeled data were accurately classified with no cumbersome annotation of data performed. Furthermore, no data-pre-processing steps were needed, such as rescaling mean and variance of IBI values, and noise is mostly discarded in the encoding IBI values with respective quality indices. DNNs are preferably trained on raw-data, as they can extract information from data that human observers would miss; however, even with the use of pre-processed data, such approaches improve detection of AF. The wearable utilized in this study employed proprietary algorithms and only provided pre-processed data. This might impact the information content originally contained in the raw data. Especially in a medical context, it should be mandatory to perform context-related accuracy testing when using pre-processed data (see Appendix A). Testing the pre-processed data revealed a comparable correlation for ECG and PPG derived heart rate estimation [21]. For practical application of such medical wearables, utilization of pre-processed data may represent the more frequent use case.

In this trial, the recording time was identical to the monitoring time (=time device was used by patients), driven by the fact that the wearable was attached and dismantled by the investigator. However, this might be different in daily practice, as patients might, e.g., wear the device while the battery is empty. It is of interest to note that in other studies, no clear time definitions and data are provided, e.g., in the Apple Heart and Huawei Heart study [10,12]. An analysis of variables that have an impact on interpretable time in this trial is at least partly in accordance with published data [22]. The impact of age and height on the interpretable time shown by the logistic regression was modest. 

The European Society of Cardiology guidelines on the management of AF recommends screening for silent AF with ECG-based devices in selected patient populations [4]. New technologies, such as smar watches (ECG and PPG based), are not yet recommended in the guidelines as no formal evaluation of these devices has been performed yet. Passive monitoring approaches with wrist-worn smartwatches (as those used in the Apple Heart and Huawei Heart studies) showed an acceptable diagnostic performance in a non-risk population. However, the performance of such devices is not sufficient for screening for silent AF, due to their low interpretable time with respect to a 24 h measurement. It is known that recording of ECG Holter is hampered by noisy measurements and/or artifacts induced by physical activity, thus potentially leading to under-diagnosis of AF episodes. In a recent analysis, an elimination rate of 30% (i.e., not interpretable ECG recording time) of data was observed [23]. A disadvantage of conventional adhesive ECG-patches used until now is the limited adherence of patients, due to discomfort, visibility, and skin reactions. 

The wearable used in this trial was chosen because of a tight upper arm fit in order to reduce artifacts induced by probe-tissue movement, e.g., due to physical activity [15]. In this respect, it is worth mentioning that the activity index had no significant impact on interpretable time. Moreover, ambient light emitted by external sources interference is minimized by a sensor location most often covered by clothing. The medical wearable could be connected to secure web-based services, and thus, provide immediate feedback. The respective results of the questionnaire used in this trial showed that the patients appreciated the non-invasive wearable; however, it was used for one day only. It remains to be studied if patients are willing to wear such a medical wearable for long-term monitoring (=high adherence rate) as it is not a ‘lifestyle-device’. Nevertheless, patients might favor the comfort of such a wearable in contrast to other options.

Till today it is still under discussion which duration of AF burden is associated with an increased risk for clinical complications, such as ischemic stroke [6]. Considering the commercially available wearables and the studied device, data acquisition is based on a block-wise approach (i.e., five-minute time periods). It is not clear which time resolution (=number of data points per time unit) is needed in order to be able to detect all AF episodes with sufficient diagnostic accuracy. 

In summary, medical wearables with such specifications offer the option of permanent surveillance, i.e., live monitoring of the patients by health care professionals. The workload of specialized clinics may be reduced if live remote patient monitoring was enabled by modern wearables. This could be a contributively brick for structured disease management applications [24]. 

This trial evaluated only hospitalized patients at high risk for AF in a proof of concept approach. Sensitivity and specificity have to be further evaluated in a population at a lower risk for AF. For this study, we evaluated a population with a high risk of AF. Importantly, patients at ‘moderate’-risk of AF might represent the most relevant population, in whom longer monitoring times are required to detect AF episodes. Compliance and adherence were only tested in patients carrying the wearable for 24 h. It remains to be studied how good the acceptance of the wearable is over longer time periods. Such studies would also help to see how limits of the current version of this wearable can be handled. If these and other exogenous factors can be overcome, this would achieve a high interpretable time to maximize high-resolution data. A limitation of our study was that we had to rely on data acquisition and raw data analysis that was implemented in the wearable and on proprietary quality indices. It is acknowledged that the signal quality index is critical for AF detection, as noisy sinus rhythm might be mis-detected as AF. A preliminary accuracy testing was performed (see Appendix B).

## 5. Conclusions

In conclusion, detection of AF with a medical wearable attached to the upper arm is a feasible and reliable approach, also during physical activity for remote monitoring purposes. The results presented encourage the performance of long-term clinical trials with a focus on everyday conditions. Assuming a positive outcome of such studies, monitoring of patients with AF might move away from Holter ECG towards medical wearables. 

## Figures and Tables

**Figure 1 sensors-20-05517-f001:**
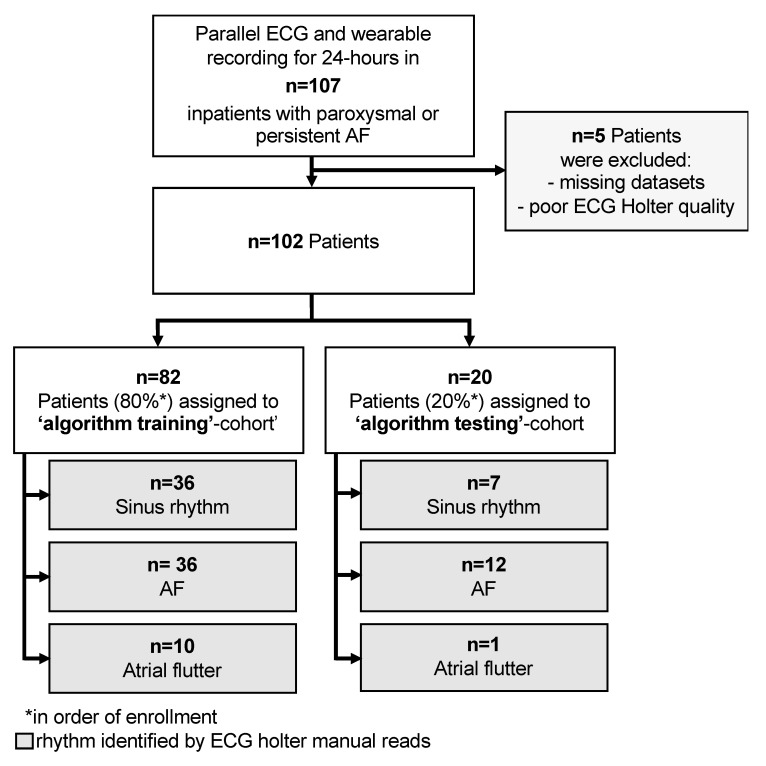
Flow-chart of patient disposition for algorithm development and group classification for the trial.

**Figure 2 sensors-20-05517-f002:**
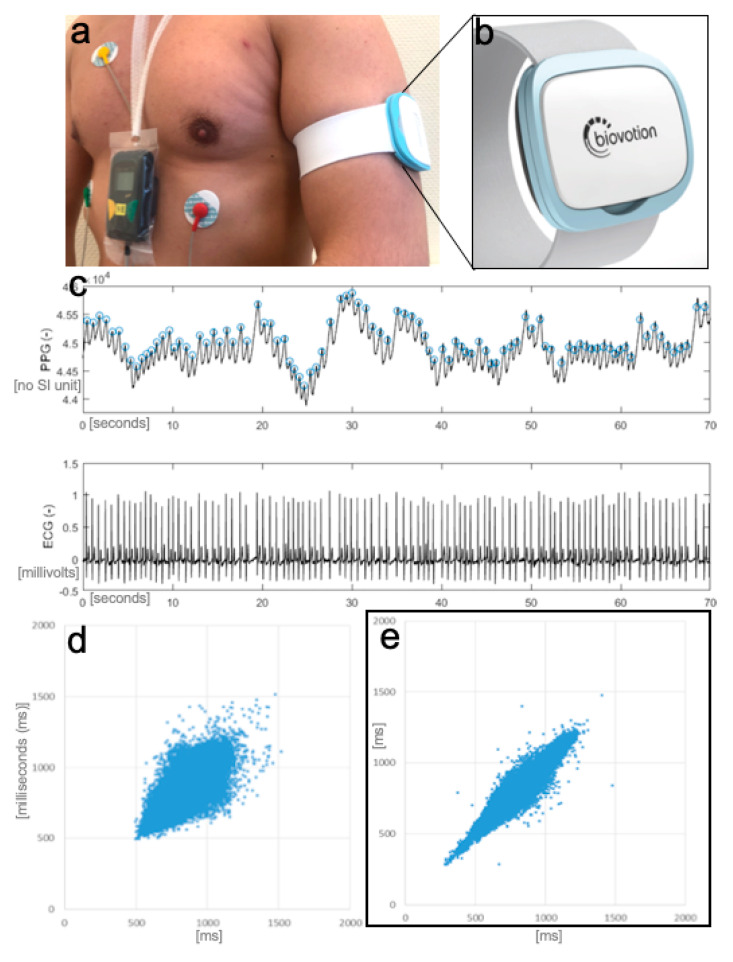
Recordings setup with the medical wearable attached to the left upper arm and ECG Holter (**a**); wearable (**b**); recorded signals (**c**) (first-row photoplethysmography (PPG), second-row ECG; showing an atrial fibrillation (AF) recording); Poincaré plot of PPG derived inter-beat intervals in AF (**d**) and sinus rhythm (**e**).

**Figure 3 sensors-20-05517-f003:**
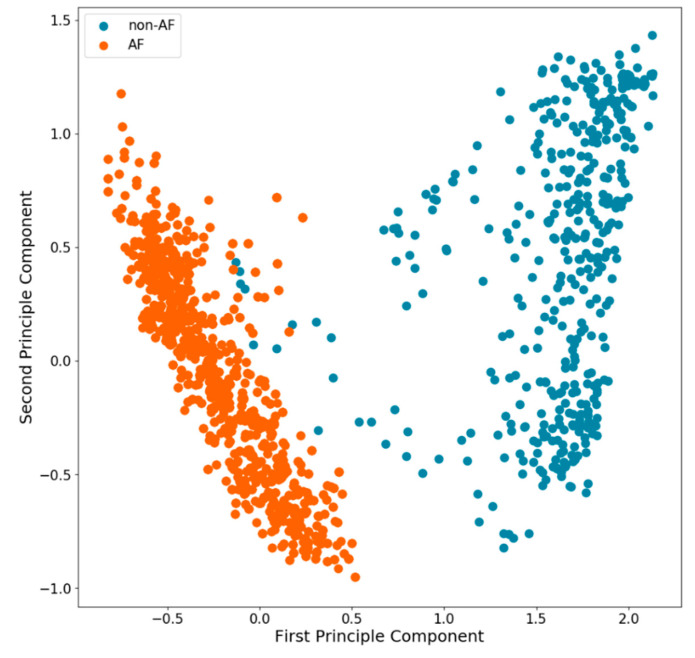
First two principle components of the latent space from the unsupervised Deep Learning approach for five-minute periods. Results of one-nearest neighbor classification for individual periods are shown that would be interpreted as AF (orange) or non-AF (blue).

**Figure 4 sensors-20-05517-f004:**
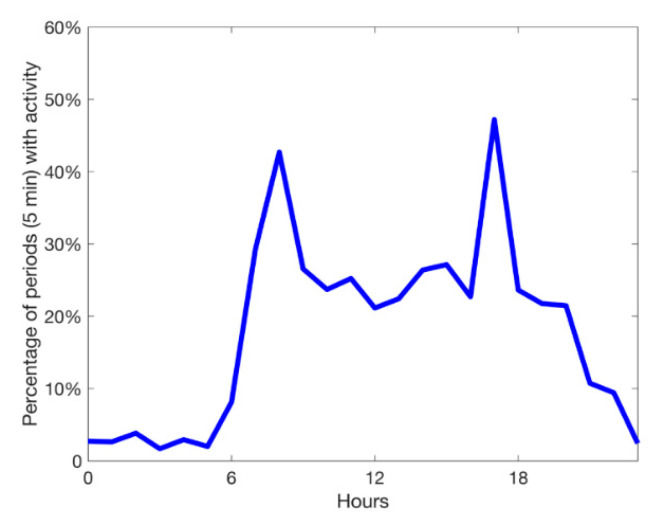
Percentage of five-minute periods with a positive physical activity index of all patients for 24 h.

**Table 1 sensors-20-05517-t001:** Demographics, comorbidities, concomitant medication, and CHA_2_DS_2_-VASc Score of patients enrolled.

Patient Characteristics	No. (%)
Sex	
Male	53 (52.0)
Female	49 (48.0)
Age [years]	71.0 ± 11.9
Height [cm]	176.6 ± 10.8
Weight [kg]	86.1 ± 20.0
BMI [kg/m^2^]	28.8 ± 5.4
Arm circumference [cm]	29.6 ± 3.7
Comorbidities	
Arterial hypertension	82 (80.4)
Diabetes mellitus	20 (19.6)
Stroke/ Myocardial infarction	21 (20.6)
Reduced left ventricular ejection fraction	32 (31.4)
Peripheral vascular disease	2 (1.9)
CHA_2_DS_2_-VASc-Scores	
0	8 (7.8)
1	15 (14.7)
2	17 (16.7)
3	30 (29.4)
4	23 (22.5)
5	9 (8.8)
>5	0 (0.0)
Mean	2.7 ± 1.4
Concomitant medication	
Anticoagulants	90 (88.2)
Antiplatelet	14 (13.7)
Beta-blocker	82 (80.4)
Calcium channel blocker	23 (22.5)
Renin-angiotensin system inhibitors	68 (66.7)
Other antihypertensive drugs	52 (51.0)
Other antiarrhythmic drugs	16 (15.7)
Glycosides	9 (8.8)
Heart rhythm by ECG Holter reads	
Sinus rhythm	43 (42.2)
Atrial fibrillation	48 (47.0)
Atrial flutter	11 (10.8)

**Table 2 sensors-20-05517-t002:** Mean interpretable time, sensitivity, specificity, positive predictive value, negative predictive value, and AUC of ROC-analysis for detection of AF by using PPG analysis overall and during moderate physical activity and the average sensitivity/ specificity with SD estimated with 1-nearest neighbor classification and a deep neural network trained on five-minute periods on different training and validation test splits.

Method	Sensitivity [%]	Specificity [%]	PPV [%]	NPV [%]	AUC
nRMSSD -periods in physical activity	95.2 92.9	92.5 85.5	70.1 63.1	97.8 97.7	0.97 -
1-nearest neighbor classification -periods in physical activity	96.0 ± 0.4 96.8 ± 0.6	99.0 ± 0.2 96.9 ± 0.5	94.7 ± 0.6 94.3 ± 0.4	99.3 ± 0.0 99.3 ± 0.1	0.98 ± 0.2 -
DNN (classifier trained on annotated data) -periods in physical activity	97.0 ± 0.3 97.0 ± 0.3	95.0 ± 0.4 95.8 ± 0.4	81.0 ± 1.3 83.8 ± 1.2	99.3 ± 0.1 99.3 ± 0.1	0.99 ± 0.2 -

nRMSSD = normalized root mean square of the successive difference, DNN = deep neural network, PPV = positive predictive value, NPV = negative predictive value, AUC = area under the receiver operating characteristics curve, ROC = receiver operating characteristic.

**Table 3 sensors-20-05517-t003:** Differences in demographics, medical characteristics, concomitant medication, and measurement conditions (below the bold line) of patients with interpretable time < 80% and ≥ 80%. (Significant differences are marked in bold, Continuous variables are given as mean ± SD).

Characteristics	Interpretable Time < 80%	Interpretable Time ≥ 80%	*p* Value
	No. (%)		
Count	40 (39.2)	62 (60.8)	
Sex Male Female	18 (45.0) 22 (55.0)	35 (56.5) 27 (43.5)	0.312
Age [years]	74.3 ± 9.8	68.9 ± 12.8	0.023
Height [cm]	168.6 ± 10.2	175.1 ± 10.6	0.003
Weight [kg]	86.8 ± 23.5	85.7 ± 17.6	0.619
Arterial hypertension	35 (87.5)	47 (75.8)	0.203
Diabetes mellitus	9 (22.5)	11 (17.7)	0.614
Stroke/myocardial infarction	10 (25.0)	11 (17.7)	0.454
Reduced left ventricular ejection fraction	17 (42.5)	15 (24.2)	0.080
Peripheral vascular disease	1 (2.5)	1 (1.6)	nA
CHA_2_DS_2_-VASc-Scores 0 1 2 3 4 5	2 (5.0) 2 (5.0) 6 (15.0) 13 (32.5) 12 (30.0) 5 (12.5)	6 (9.7) 13 (21.0) 11 (17.7) 17 (27.4) 11 (17.7) 4 (6.5)	0.172
Anticoagulants	36 (90.0)	54 (87.1)	0.760
Antiplatelet	4 (10.0)	10 (16.1)	0.557
Beta-blocker	37 (92.5)	45 (72.6)	0.020
Calcium channel blocker	13 (32.5)	10 (16.1)	0.088
Renin-angiotensin system inhibitors	32 (80.0)	36 (58.1)	0.031
Heart rhythm Sinus rhythm Atrial fibrillation Atrial flutter	14 (35.0) 23 (57.5) 3 (7.5)	29 (46.8) 25 (40.3) 8 (12.9)	0.225
Arm circumference [cm]	29.9 ± 2.9	29.9 ± 4.7	0.559
Activity index (median)	14.7%	14.9%	0.204

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
