# Peer review of "Reliable Detection of Atrial Fibrillation with a Medical Wearable during Inpatient Conditions"

_sensors, 2020, doi:10.3390/s20195517_

Round 1

Reviewer 1 Report

Overall comments:

This manuscript was to evaluate sensitivity and specificity of non-invasive AF detection by a medical wearable. There are several areas where the manuscript needs to be strengthened.

Specific comments:

  1. Please describe more details to calculate the power of this study.
  2. Describe any efforts to address potential sources of bias, such as how to deal with the procedure if applicable.
  3. It will be better to show kappa for the two cardiologists independently using a standard of care software tool.
  4. P-values typically only need to be reported to 1 significant figure.
  5. What is the originality and strengths of this case report? How physicians or policy makers could deliberate with patients or people based on the key findings of this manuscript?
  6. The authors should add the comments related to selection bias in this study to the perceived limitation subsection.

Totally, I would like to congratulate the authors for the enthusiasm invested in this study.However, the manuscript does not reach the level of quality required for publication as original research without some revisions in Sensors.

Reviewer 2 Report

Dear Authors,

I would first congratulate you for you work! The study you have performed is really valuable from clinical and scientific point of view. 

I have just a few recommendations and questions, that need clarification:

Line 42: “Ischemic stroke” sounds better than “Major stroke”

Lines 78-80: A primary endpoint is actually the main result that is measured at the end of a study to see if a given treatment worked (e.g., the number of deaths or the difference in survival between the treatment group and the control group). Please, clarify/modify this part of the text…

Line 81-82: About the inclusion criteria: The true value of these devices would be to detect patients with moderate risk of AF, many of them requiring longer monitoring period. High risk individuals such as very elderly patients, those with valvular heart disease, very dilated left atrium, etc. would probably be diagnosed quite well by 24 h conventional ECG-Holters. Modern Holter-devices are small-sized, precise enough and not so uncomfortable to the patients...

In addition, since you mention about high-risk patients for AF, would you specify how you evaluated (the algorithm for) the risk/probability of AF occurrence/recurrence?

Reviewer 3 Report

The present paper aims to assess sensitivity and specificity of a wearable photoplethysmography medical device compared to ECG Holter in detecting paroxysmal or persistent atrial fibrillation at rest and during moderate physical activity.

A few changes are needed, as follows:

Introduction: Please add a few words about risk factors of atrial fibrillation. Modifiable cardiovascular risk factors including obesity, hypertension, diabetes mellitus, obstructive sleep apnea, alcohol consumption, smoking, and sedentary lifestyles are possible contributors to the development and progression of AF (Shamloo et al. Atrial fibrillation: A review of modifiable risk factors and preventive strategies. Rom J Intern Med. 2019;57(2):99-109. doi: 10.2478/rjim-2018-0045; Mozos I. Arrhythmia risk and obesity. J Mol Genet Med. 2014:S1: 1747-0862). Please mention also that electrophysiological remodeling and oxidative stress are involved in the pathogenesis of AF (Gasparova et al. Perspectives and challenges of antioxidant therapy for atrial fibrillation. Naunyn Schmiedebergs Arch Pharmacol. 2017 Jan;390(1):1-14).

Material and methods: How did you evaluate risk for AF?

Table 1: Please calculate also BMI considering that obesity is frequently associated with AF (Mozos I. Arrhythmia risk and obesity. J Mol Genet Med. 2014:S1: 1747-0862). Please include also BMI in table 3!

Table 2 requires a footnote to explain all abbreviations.

Reviewer 4 Report

General comments:

The authors of the current study presented two algorithms for non-invasive detection of AF using a commercial wearable PPG monitor placed on the upper arm of patients. The target population were patients with higher risk of developing episodes of AF, which wore the monitor continuously for a period of 24h, in conjunction with a control in the form of an ECG Holter. The performance results obtained by the authors have shown AF sensitivities of 95.2% and 97% and specificities of 92.5% and 95% for the nRMSSD and DNN algorithms, respectively. The authors also claim that AF monitoring was feasible during moderate exercise performed by the patients, though the tested population only had a typical ratio of 16% of time spent on activity engagement in a 24h period. In term of the human trial, it is a well conducted study where authors thoroughly describe the history of the patients (demographics, comorbidities, medication and score) and their possible relation to AF, given the different interpretable times of the recordings. However, as the authors have relied on a commercial wearable device with property software (as far I could understand as reviewer of the manuscript), this diminish the impact of the current study in terms of novel technologies for AF monitoring. I kindly ask the authors to comment on the topics below, which I believe can improve the quality of the manuscript.

Technical comments:

1) Abstract: please define AUC when used for the first time in the written text.

2) Introduction:

. Line 41: there is no need of abbreviating AF again.

. Line 47: “ECG Holter’s” replace by “ECG Holters”

. Line 49: What does EC stand for?

. Line 51: the term photo plethysmography. Are the authors sure that the correct term is “photo”? Should it not refer to “pulse” instead?    

. Line 52: Consider also to describe more about the passive approach, as done for the active one.

. Lines 54 – 55: This sentence is not understandable. Please consider rephrasing it.

. Overall: the introduction is too short as it stands right now. Authors should, for instance, discuss more about AF and its incidence on the world’s population (percentage), adding also more literature references to this topic. More comparison text with other PPG systems is required, namely, the conventional finger PPG. Within the same reasoning, the physiological aspect relating the bio-potential recording (ECG waveform) with the arterial pulse wave at the periphery of the cardiovascular system must be referred within the text: does the arterial pulse wave (maxima) really follow the heart’s depolarization wave (QRS complex) and so, the heart beat can trustworthy be monitored by PPG, especially with heart diseases involved (like AF)? A reference to the upper arm wearable device is made, but no description about the technical details is provided, namely, sample rate, signal resolution, data logging, elapsed time between pulse excitation/acquisition and data memory storage, capacity of device storage, etc.

3) Material and Methods:

. Line 79: please define more technically “moderate physical activity”. What type of exercises does it encompass?

. Figure 1: define the technical conditions that led to the exclusion of different datasets. How was the quality of ECG (or PPG) signal assessed?

. Figure 2: please consider adding labels for the x and y axes in each graphic presented. Moreover, the measurement units in the y-axis must be specified for PPG and ECG.

. Line 111: elaborate more on the “insufficient quality” criterium.

. Line 123: the characteristics of the AF data oversampling are missing.

4) Results:

. Lines 181 – 182: reason for the large difference between periods of classified AF in the wearable data (4469) and Holter (5156)? The authors should comment on this discrepancy considering the different type of transduction mechanisms involved to record the heartbeat and mechanical fixation of both systems to the surface of the body. Is the upper arm the best place to record the pulse through PPG?

. Table II needs to be reformatted to distinguish between metrics for AF obtained by the wearable PPG and Holter system separately and also divided into rest and moderate physical activity for better readability and interpretation of the results.

5) Discussion:

. Line 254: “Brasier et al.” replace by the reference number only or add it after the name of the authors, so readers can access their study directly and not search for the first author’s name.

. Line 262: add the reference number for the Apple Heart study.

. Line 270: again correct for “Tison et al.”

. Line 273: “noise is mostly discarded in the encoding IBI values”. How can the authors ensure this if they did not develop the hardware (wearable device)? Is there a metric provided by the manufacturer to assess the performance of the readings under noisy environments? If so, please add it to the written text.

. Lines 276 – 277: this sentence is not entirely understandable. Please rephrase it.

. Line 303: “In this respect it worth” replace by “In this respect it is worth”

6) Appendix B:

Line 425: “(van den Oord et al)” please add the respective reference number.

Round 2

Reviewer 4 Report

General comments:

I congratulate the authors for the modifications made to the manuscript in order to increase its quality and readability. But, before complete acceptance, I ask the authors to review the following points:

1) Lines 108-109 (Materials and Methods): Should it not be:” ECG datasets were discarded if less than 50% of recorded data was NOT interpretable, as defined by our independent raters”?

2) Figure 2: Time units in the graphic C? Seconds? The [arbitrary] units in the PPG graphic are confusing. If the authors do not have access to the specifications of the manufacturer (biovotion), it is more correct to write on the figure caption that “the units for the PPG device were not provided by the manufacturer”, therefore removing the [arbitrary] from Y-axis. As reviewer in electronics, it seems to me that the values showed by the wearable device are in the digital range of 16-bit (0 – 65535), which is the expectable resolution for light signal acquisition involved in PPG.

3) Line 450 (Appendix B): Reference [12] must be replaced by reference [17] (van den Oord).

4) References section: reference 21 is not referred within the written text of the manuscript. Please correct.
